# Group Diffusion Transformers are Unsupervised Multitask Learners

## Abstract

While large language models (LLMs) have revolutionized natural language processing with their task-agnostic capabilities, visual generation tasks such as image translation, style transfer, and character customization still rely heavily on supervised, task-specific datasets. In this work, we introduce **Group Diffusion Transformers (GDTs)**, a novel framework that unifies diverse visual generation tasks by redefining them as a **group generation** problem. In this approach, a set of related images is generated simultaneously, optionally conditioned on a subset of the group. GDTs build upon diffusion transformers with minimal architectural modifications by concatenating self-attention tokens across images. This allows the model to implicitly capture cross-image relationships (*e.g.*, identities, styles, layouts, surroundings, textures, and color schemes) through caption-based correlations. Our design enables scalable, unsupervised, and task-agnostic pretraining using extensive collections of image groups sourced from multimodal internet articles, image galleries, and video frames. We evaluate GDTs on a comprehensive benchmark featuring over 200 instructions across 30 distinct visual generation tasks, including picture book creation, font design, style transfer, sketching, colorization, drawing sequence generation, and character customization. Our models achieve competitive **zero-shot** performance without any additional fine-tuning or gradient updates. Furthermore, ablation studies confirm the effectiveness of key components such as data scaling, group size, and model design. These results demonstrate the potential of GDTs as scalable, general-purpose visual generation systems. We will release the code and models to support further research.

## 1 Introduction

The advent of large language models (LLMs) has brought a paradigm shift in natural language processing (NLP) Radford et al. (2019); Raffel et al. (2020); Brown (2020); Ouyang et al. (2022); Zhang et al. (2022); Touvron et al. (2023a;b); Dubey et al. (2024), enabling a wide range of tasks to be approached in a task-agnostic manner. These models, trained on vast corpora, can generate coherent and contextually relevant content across various domains without the need for task-specific fine-tuning, setting a new standard for what is achievable in NLP. However, this level of task generalization has yet to be fully realized in the field of visual generation. Unlike NLP, visual generation tasks – such as pose transfer Shen et al. (2023); Lu et al. (2024), image translation Ho et al. (2024); Rodatz et al. (2024), customization Jones et al. (2024); Wei et al. (2023), stylization Huang et al. (2024); Yang et al. (2023), and font creation Wang et al. (2023a); Yang et al. (2024) – remain largely siloed, relying heavily on supervised learning paradigms. These tasks often demand extensive task-specific datasets and additional modules, such as LoRAs Jones et al. (2024); Smith et al. (2023); Luo et al. (2023), adapters Ye et al. (2023a); Mou et al. (2024), visual encoders Giannone et al. (2022); Kumar et al. (2024); Xu et al. (2024), and ControlNets Zhang et al. (2023); Zhao et al. (2024), to achieve satisfactory performance.

This reliance on specialized data and architectures presents significant challenges for scalability and generalization. First, it limits scalability by failing to leverage the vast amount of weakly supervised data available on the Internet; creating and curating task-specific datasets is human-laboring. Second, it restricts models' adaptability to unseen tasks. Third, cross-task adaptation is lacking, particularly in compositional control, where multiple tasks are implicitly managed. For example, consider creating a picture book Jin & Song (2023); Wang et al. (2023b), characters, environments,

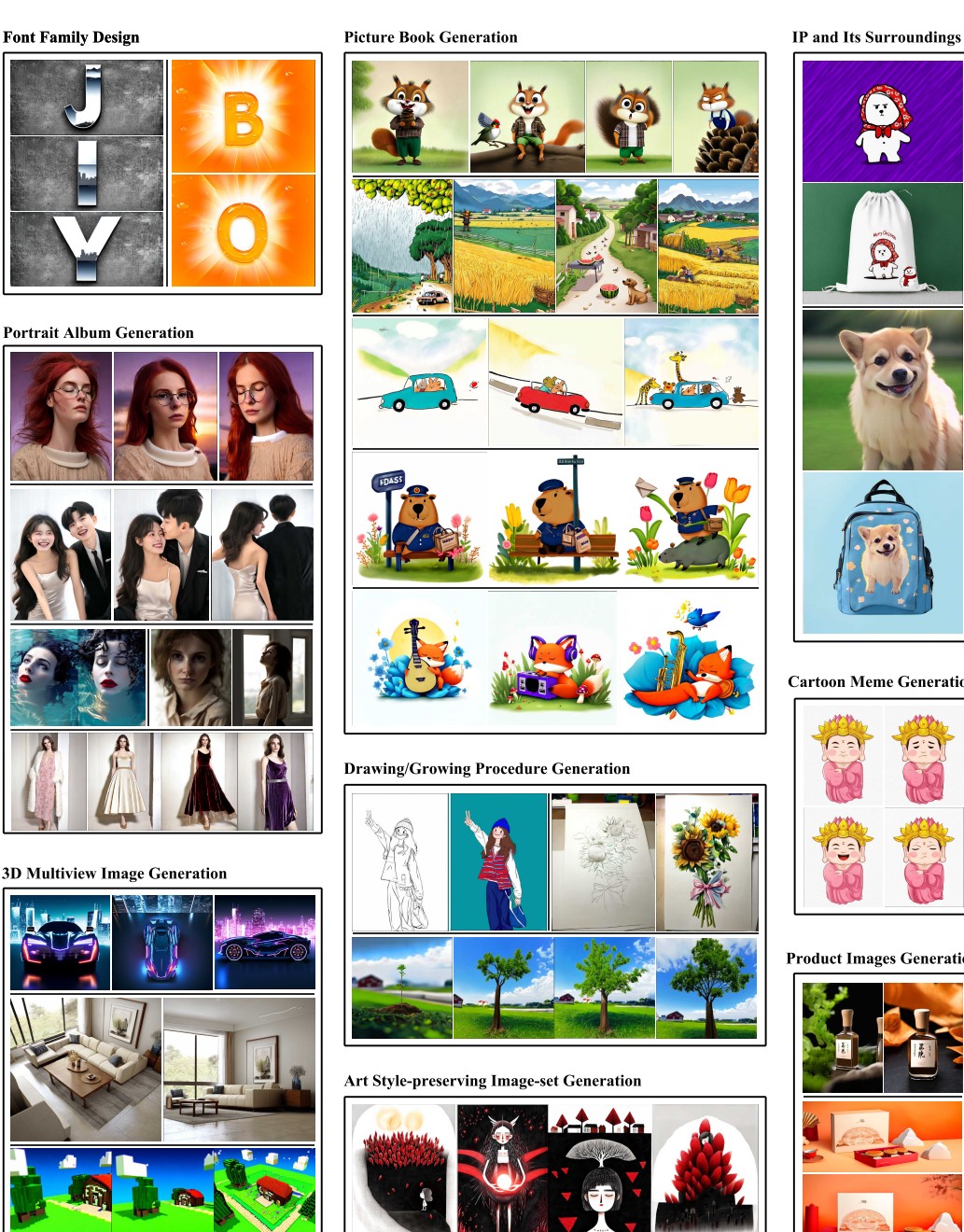

Figure 1: **Group Diffusion Transformers perform a vast array of visual generation tasks in a unified framework termed group generation.** Note that **NO** task-specific dataset and **NO** additional gradient update is applied. The model is automatically generalized to these tasks after unsupervised training on image groups. For simplicity, textual descriptions of images are omitted here, which can be found in Appendix.

and attire must be dynamically adjusted, requiring decisions on which elements to keep consistent and which to vary. Finally, we hypothesize that training on single-task, shallow-domain datasets leads to the lack of generalization in real-world applications. To truly unlock the potential of visual generation, it is crucial to develop models capable of performing a wide range of tasks in a task-agnostic manner. This demands a shift in how we conceptualize and approach these tasks.

Our key insight is that most, *if not all*, visual generation tasks can be reformulated within a unified framework that we term the **group generation** problem. In this framework, the objective is

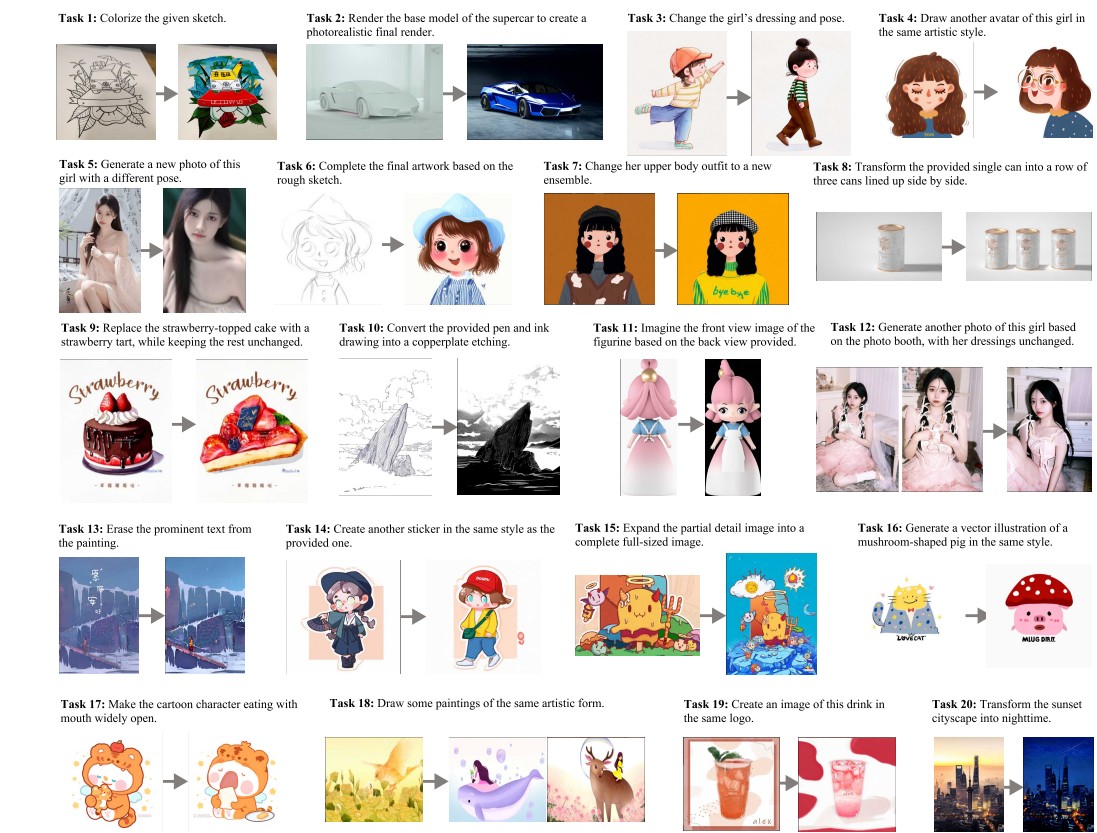

Figure 2: **When conditioned on a subset of the group data, Group Diffusion Transformers could perform conditional group generation in the inpainting fashion.** Note that the model is automatically generalized to these tasks after unsupervised training on image groups. Textual descriptions of images are omitted here (can be found in Appendix), and we summarize them into brief task descriptions.

to generate a set of correlated data, or a *group*, optionally conditioned on a subset of this group. For instance, tasks such as generating picture books Jin & Song (2023); Wang et al. (2023b), font images Wang et al. (2023a); Yang et al. (2024), or emoticons Mittal et al. (2020) involve producing multiple images with distinct yet related descriptions simultaneously. The inherent correlations are implicitly captured through the relationships among these descriptions. Conversely, tasks like sketching Voynov et al. (2023); Wang et al. (2023c), colorization Zabari et al. (2023); Carrillo et al. (2023); Liang et al. (2024), character-specific image generation Zdenek & Nakayama (2023); Kou et al. (2023), and multiview image generation from a single image Liu et al. (2023b); Shi et al. (2023) can be framed as conditional group generation problems, where a subset of the group data is provided as a reference. Figure 1 and 2 provide examples of group generation and conditional group generation. By reframing these tasks as group generation problems, we leverage the power of unsupervised learning to address a broad spectrum of tasks without the need for task-specific supervision, simplifying the learning process and broadening applicability.

One of the most compelling advantages of the **group generation** framework is its natural alignment with the vast amount of data available on the Internet. Multimodal articles, image galleries, and multi-shot videos are just a few examples of readily accessible sources of group data. Each of these sources inherently captures the relationships between different data elements, offering a form of free supervision that is both scalable and diverse. The availability of such abundant group data not only reduces the need for labor-intensive data annotation but also enables the training of models on a wide array of tasks simultaneously, further enhancing generalizability.

To address the group generation problem, we introduce a minimalistic modification to diffusion transformers Peebles & Xie (2023); Esser et al. (2024a); Chen et al. (2023a), termed **Group Diffusion Transformers (GDTs)**. The core idea is to concatenate self-attention tokens across a group

of inputs, allowing the model to learn the correlations and variations within the group. This modification is straightforward, requiring minimal changes to the underlying architecture of diffusion transformers (DiTs), yet it significantly enhances the model's ability to capture relationships among multiple generated data. To address reference-based generation problems, such as style transfer Huang et al. (2024); Yang et al. (2023) and image translation Ho et al. (2024); Rodatz et al. (2024), we incorporate techniques like SDEdit Meng et al. (2021) and inpainting Xie et al. (2023); Xu et al. (2024). These methods enable the model to generate the remaining elements of a group when conditioned on a subset of inputs. Figure 3 provides a detailed architectural overview of GDTs. The straightforward design of GDTs makes it easy to implement and shows promise for efficient scaling.

To evaluate the capabilities of our model, we first introduce a user interface that can automatically convert user instructions into textual descriptions of the target image group to support group generation. Then, we construct a comprehensive benchmark that covers a wide range of visual generation tasks, both with and without reference images. All tasks are performed in a zero-shot setting, without any parameter or architectural modifications. Despite the absence of task-specific supervision during training, our model demonstrates promising performance across most tasks. Finally, we conduct ablation studies to examine the impact of key components in our framework, such as data scale, group size, model design and quality tuning, on overall performance.

## 2 APPROACH

The core of our approach is to reformulate visual generation tasks into a *group generation* problem and solve it using minimally modified diffusion transformers. We begin by detailing how these tasks are reformulated, followed by a comprehensive introduction to our model, its architecture, the data employed, the training procedure, and the inference stage.

### 2.1 PROBLEM FORMULATION

We propose that a vast array of visual generation tasks can be unified under a single framework we term the **group generation** problem. In this framework, the objective is to generate a group of $n$ elements $\mathbf{x} = \{\mathbf{x}_1, \mathbf{x}_2, \cdots, \mathbf{x}_n\}$, where each element is conditioned on its respective context (*e.g., image descriptions*) $\mathbf{c} = \{\mathbf{c}_1, \mathbf{c}_2, \cdots, \mathbf{c}_n\}$. The relationships among these elements are implicitly defined by the interdependencies within their contextual conditions. Optionally, a subset of $0 \leq m < n$ elements of $\mathbf{x}$ can be provided as reference data, with the task being to generate the remaining $(n - m)$ elements. This formulation naturally encapsulates a variety of tasks:

- **Text-to-Image:** A special case where the group size $n = 1$ and the reference subset size $m = 0$. The task is to generate a single image from a textual description.
- **Font Generation:** Here, the group size $n > 1$ corresponds to the number of characters to generate, with $m = 0$.
- **Picture Book Generation:** Similar to font generation, the group size $n > 1$ corresponds to the number of picture book pages, with $m = 0$. The descriptions capture the connections and variations across the pages.
- **Identity Preservation:** Here, the group size $n > 1$ corresponds to the number of photos with the same identities to generate, with $m = 0$. Identity-specific information is reflected in the descriptions, such as names or other identifiers.
- **Local Editing:** In this task, the group size is $n = 2$ with a reference subset size $m = 1$. One reference image is provided, and the model generates the edited image based on the differences captured in their descriptions.
- **Image Translation:** Similarly, the group size is $n = 2$ with a reference subset size $m = 1$. A reference image from one domain is converted to another domain according to their descriptions.
- **Subject Customization:** The task involves generating $(n - m) \geq 1$ images, where $1 \leq m < n$ character images are used as references.
- **Style Adaptation:** In this task, $(n - m) \geq 1$ corresponds to the number of stylized images to be generated, with $m = 1$ being the reference image guiding the target style.

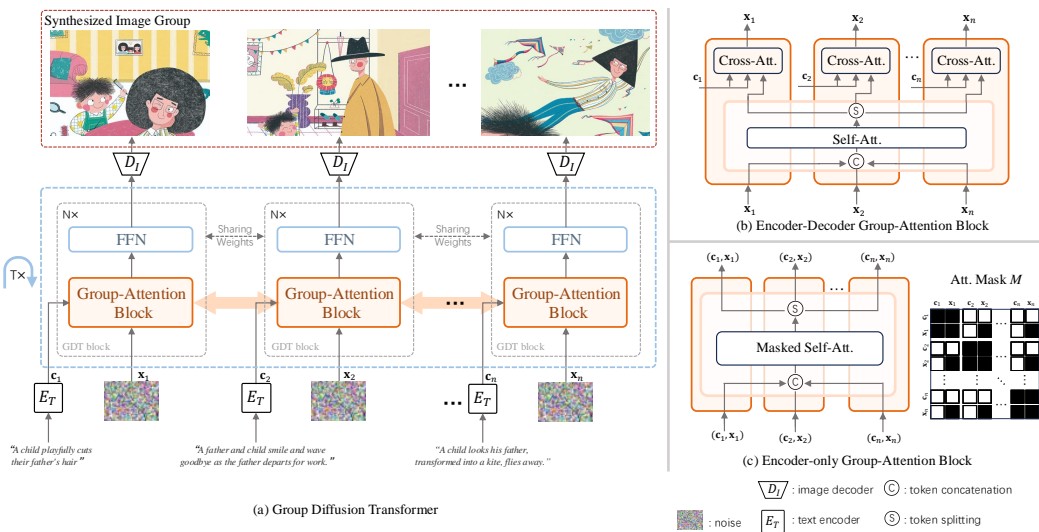

Figure 3: **The overview of Group Diffusion Transformer, which takes minimal adaptations for the encoder-decoder and encoder-only visual generation architectures.** We make a straightforward modification on self-attention blocks by concatenating image tokens across group inputs, allowing to learn inter-image correlations.

These examples illustrate just a few of the many tasks that can be naturally expressed within the *group generation* framework. Across these tasks, the task hints are naturally embedded within the group element descriptions, much like how a human might communicate with a designer. This unified framework simplifies the approach to diverse visual generation tasks and paves the way for scalable, generalized solutions.

## 2.2 MODEL AND ARCHITECTURES

To tackle the group generation problem, it is crucial to establish connections between multiple group elements during the generation process, allowing the model to perceive and utilize the correlations among these elements. Our approach involves a straightforward modification: concatenating tokens across group inputs within the self-attention blocks of diffusion transformers. This enables tokens from different data elements to interact with one another throughout the model's layers.

For different text-conditioned visual generation architectures, we make minimal adaptations to accommodate our approach:

- **Encoder-Decoder:** In architectures like PixArt Chen et al. (2023a), each transformer block includes a self-attention operation for the image, cross-attention for interaction between image and text, and a feed-forward network. We choose to concatenate all the image tokens in self-attention blocks, which allows every token attends to all the image tokens within the group. After self-attention operation, concatenated image tokens are split correspondingly. Then, in cross-attention blocks, each image token attends only to the text embeddings associated with its respective description. This setup is illustrated in Figure 3 (b).

- **Encoder-Only:** Examples like Stable Diffusion 3 Esser et al. (2024a) and FLUX Labs (2024) feature transformer blocks with self-attention blocks and feed-forward networks. We modify the self-attention operation into a masked version, which is depicted in Figure 3 (c). Specifically, image tokens $\mathbf{x}_i$ as well as text tokens $\mathbf{c}_i$ are first concatenated with each other all over the group. Then, we calculate the masked self-attention, where the mask is designed for allowing every image token attends to all tokens across the group while allowing context tokens only attend to image tokens as well as themselves. Concretely, let $M(\mathbf{a}_j, \mathbf{b}_k)$ indicate the attention mask for tokens in $\mathbf{a}_j$ and $\mathbf{b}_k$, where $\mathbf{a}, \mathbf{b} \in \{\mathbf{c}, \mathbf{x}\}, 0 \leq j, k \leq n$. Then, $M(\mathbf{a}_j, \mathbf{b}_k)$ is decided by

$$M(\mathbf{a}_j, \mathbf{b}_k) = \begin{cases} 1 & \text{if } (j = k) \text{ or } (\mathbf{a} \in \mathbf{x} \text{ and } \mathbf{b} \in \mathbf{x}) \\ 0 & \text{else} \end{cases}. \tag{1}$$

## 2.3 TRAINING DATASET

We focus on image-related tasks in this work, which requires a high-quality, large-scale, and diverse image group dataset. While existing multimodal datasets like MINT-1T Awadalla et al. (2024) are large, they fall short of our pretraining needs due to low image quality and biased group type distribution relative to real-world visual generation applications. Thus, we construct our own dataset by sourcing image groups from multimodal Internet articles.

Our dataset creation process involve several key steps: (1) We collect a substantial amount of multimodal data, extracting images while preserving their original order to maintain group integrity. (2) A small subset of these image groups is manually annotated as either positive (suitable for retention) or negative (to be discarded). (3) Using these annotations, we train a binary classifier to score and filter the collected image groups. (4) We perform deduplication across and within groups to eliminate redundant groups and images. After preprocessing, we compile a dataset of approximately 500,000 image groups, with the distribution of group size illustrated in Figure 4.

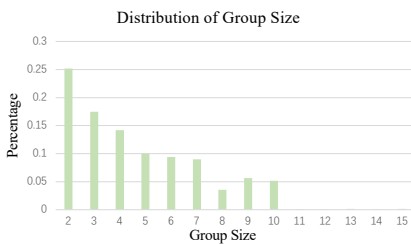

Figure 4: **Distribution of group size in our training dataset.**

The next crucial step is to generate descriptions that accurately capture the correlations among the images within each group. To achieve this, we utilize our internal multimodal large language models (MLLMs), iteratively testing and refining prompts to ensure the generated descriptions are stable and applicable across different group types. In Figure 5, we show the prompt we used, as well as the resulting group image descriptions.

While pretraining on our large-scale dataset provides a solid foundation for learning correlations with Group Diffusion Transformers (GDTs), it is common practice in visual generation tasks to conduct a supervised fine-tuning stage to enhance generation details and aesthetics. To this end, we curate a smaller, high-quality subset of approximately 10,000 image groups. These groups were selected for their strong correlations, high image quality, aesthetic appeal, and diversity. Fine-tuning our pretrained models on this curated dataset significantly improves both the image quality and content consistency in group generation, where the comparison can be found in Section 4.2.5.

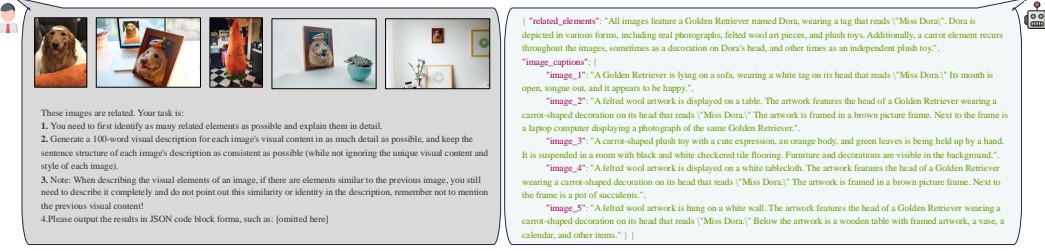

Figure 5: **Example of our training dataset, where the group images are captioned through prompting our internal MLLMs.**

## 2.4 TRAINING PROCESS

We initialize the Group Diffusion Transformers (GDTs) with weights from pre-trained text-to-image models, such as PixArt-$\alpha$ Chen et al. (2023a) and Stable Diffusion 3 Peebles & Xie (2023). Since GDTs introduce no additional parameter to the existing diffusion transformers, the pretrained weights are fully compatible. During both pretraining and supervised fine-tuning, we uniformly sample group sizes ranging from 1 to 4, dynamically adjusting the batch size to maintain consistent GPU memory usage. This approach ensures balanced performance across different group sizes. The model undergoes pretraining for approximately 100,000 steps, followed by fine-tuning on a curated dataset for around 5,000 steps. All training is conducted on A100 GPUs. We adopt the same hyperparameter settings as the official models in PixArt-$\alpha$ and Stable Diffusion 3.

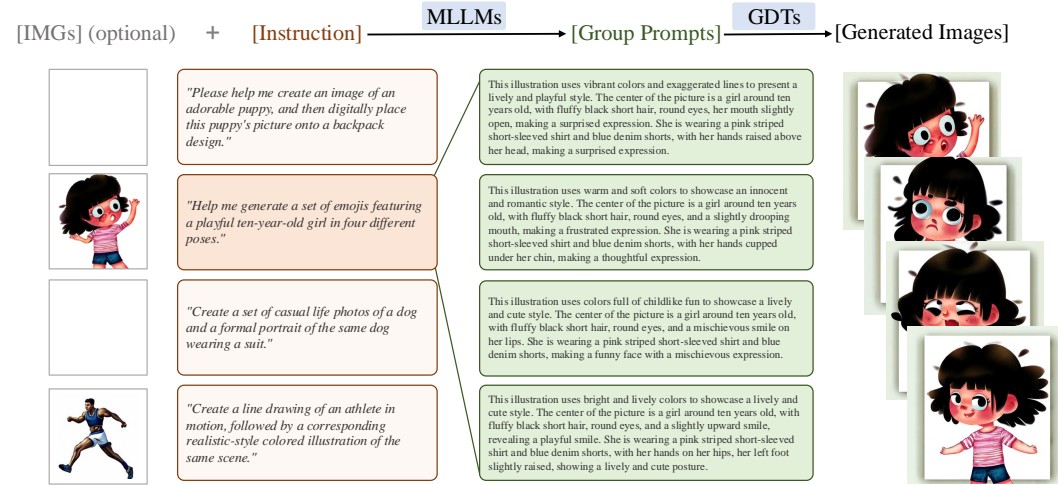

Figure 7: **We build a user interface that automatically converts the user instruction into group prompts using MLLMs, which is useful in the inference stage of GDTs.**

## 2.5 USER INTERFACE

Considering it is tedious to write a group of prompts in the inference stage, we build a **user interface** to provide a convenient interaction with the GDTs. As illustrated in Figure 7, we follow the pipeline of [`Instruction`] → [`Group Prompts`] → [`Generated Images`] for group generation, and [`IMGs`] + [`Instruction`] → [`Group Prompts`] → [`Generated Images`] for conditional group generation. Specifically, we leverage MLLMs to convert the user instruction into group prompts, where the MLLM could analyze the number of group prompts and the corresponding tasks. For example, if the instruction is "Draw a line sketch of a female character and the corresponding colored photo", the MLLM can deduce that this instruction should be transformed into two prompts, categorizing the task as sketch coloring.

## 3 BENCHMARK

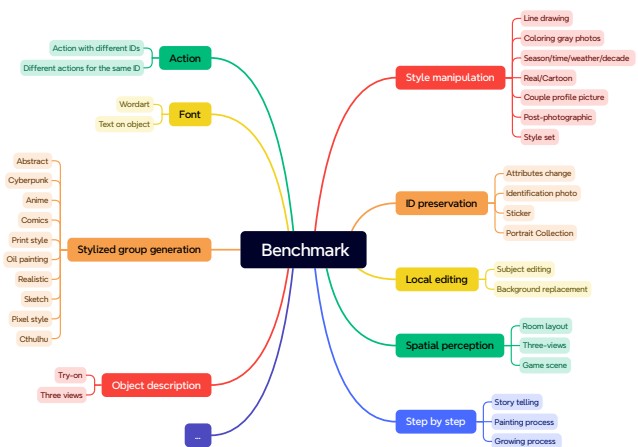

Figure 6: **Overview of our benchmark, covering about 30 distinct types of generation tasks.**

Given the diverse nature of visual generation tasks, evaluating the performance of our Group Diffusion Transformers (GDTs) presents unique challenges. Therefore, we design a benchmark that spans a wide array of tasks as shown in Figure 6. Specifically, our benchmark consists of over 200 instructions, each corresponding to one of 30 distinct types of visual generation tasks. This diversity enables a thorough assessment of the generalization capabilities of GDTs across various scenarios.

This evaluation suit encompasses tasks such as identity preservation, local editing, subject customization, font generation, and stylized group generation. Among these coarse-grained categories, further fine-grained tasks are expanded. For example, step-by-step generation contains subtasks like story telling Zhou et al. (2024), painting process Song et al. (2024), and growth process. Besides, all the textual descriptions in this benchmark are created through our user interface.

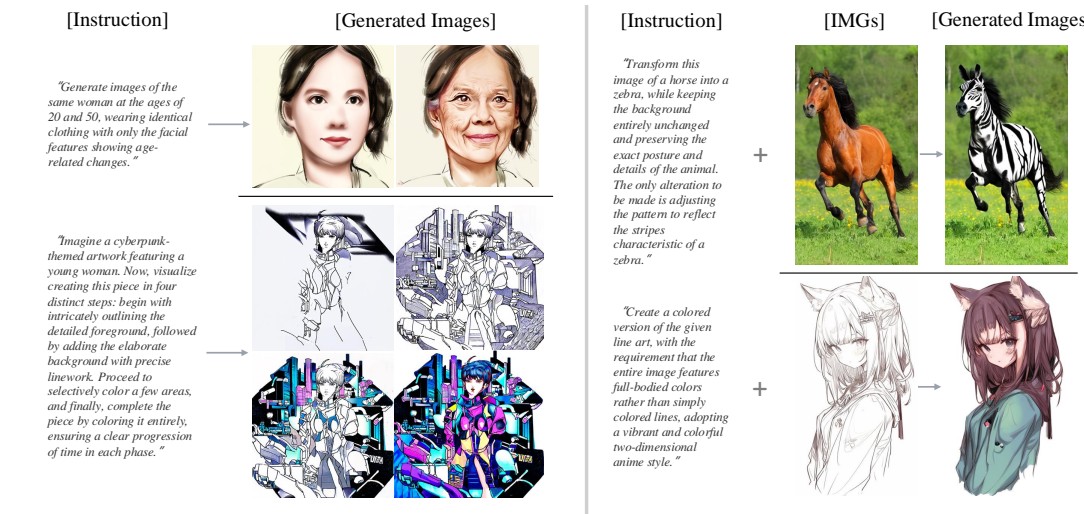

(a) Group generation       (b) Conditional group generation

Figure 8: **Generated results of GDTs on our benchmark, including group generation and conditional group generation.**

## 4 RESULTS

### 4.1 USER STUDY

We first qualitatively evaluate the generated results of GDTs on our proposed benchmark as shown in Figure 8. GDTs could perform both group generation and conditional group generation according to the user instructions. Note that the task scope of this benchmark is effectively limited by our imagination, but thanks to our unsupervised and task-agnostic pretraining, GDTs can theoretically be generalized to *arbitrary* visual generation tasks.

Table 1: **User study on our benchmark.** Human evaluation on three questions in a five-point scale.

| Models | Q1 | Q2 | Q3 |
|---|---|---|---|
| **group generation** | | | |
| PixArt-$\alpha$ | 3.44 | 3.89 | 3.78 |
| Stable Diffusion 3 | 3.20 | 3.35 | 3.29 |
| **conditional group generation** | | | |
| PixArt-$\alpha$ | 3.15 | 3.56 | 3.68 |
| Stable Diffusion 3 | 3.02 | 3.27 | 3.34 |

In our user study, we mainly adopt human ratings to assess the performance of GDTs on the benchmark. Three questions are included to measure the prompt following ability, content consistency within the image group, and the overall instruction following ability, namely: **Q1: Prompt following on each image within the group**: **Q2: Content consistency among generated group images, regardless of prompts**, **Q3: Instruction following on the generated group images.** Evaluators are asked to rate on three questions in the scale from 1 to 5, where 5 signifies perfection and 1 denotes the lowest quality. The final evaluation score is derived from the average ratings across all tasks, which serves as a robust indicator of the overall performance and its potential for real-world applications. The human-rated results are illustrated in Table 1, where GDTs achieve overall satisfaction (higher than 3) on all of the three questions.

### 4.2 ABLATION ANALYSIS

#### 4.2.1 METRICS

While our benchmark with over 200 instructions could well evaluate model's capabilities on a five-point scale, we would like to compare these ablated models in a more nuanced and quantitative manner in our ablation experiments. Therefore, we mainly present the objective metrics like FID and CLIP score. To be specific, we measure image fidelity by calculating FID on the validation set using 50k images. We assess content consistency and prompt adherence within each group by averaging CLIP similarities across every image-image and image-text pairs, respectively. In terms of reference-based generation, we adopt the same metrics but exclude pairs that involve the reference images themselves, as well as pairs between reference images and their corresponding texts.

Table 2: **Performance evaluation on key components of GDTs.** We investigate the impacts of data scale, group size, model design, and quality tuning on encoder-decoder and encoder-only models.

| Settings | PixArt-$\alpha$ (Encoder-Decoder) | | | Stable Diffusion 3 (Encoder-Only) | | |
|---|---|---|---|---|---|---|
| | FID-50k | Content Consistency | Prompt Adherence | FID-50k | Content Consistency | Prompt Adherence |
| **Data Scaling** | | | | | | |
| 5k groups | 8.40 | 0.747 | 0.291 | 8.95 | 0.740 | 0.298 |
| 50k groups | 12.06 | 0.767 | 0.293 | 10.92 | 0.760 | 0.302 |
| 500k groups | 15.91 | 0.778 | 0.300 | 11.30 | 0.761 | 0.305 |
| **Group Size** | | | | | | |
| groupsize = 2 | 15.69 | 0.784 | 0.299 | 12.37 | 0.763 | 0.301 |
| groupsize = 4 | 18.19 | 0.761 | 0.291 | 13.85 | 0.739 | 0.298 |
| groupsize = 8 | 48.26 | 0.701 | 0.252 | 18.28 | 0.701 | 0.290 |
| **Inpainting** | | | | | | |
| SDEdit | 15.71 | 0.702 | 0.299 | 12.15 | 0.751 | 0.303 |
| trainable | 10.91 | 0.725 | 0.287 | 10.94 | 0.755 | 0.298 |
| **Quality Tuning** | | | | | | |
| before | 15.91 | 0.778 | 0.300 | 11.30 | 0.761 | 0.305 |
| after | 12.53 | 0.792 | 0.298 | 10.03 | 0.781 | 0.303 |

### 4.2.2 DATA SCALING

Without the demand of task-specific supervision, it is quite easy to acquire a large abundance of group data from the Internet. We scale the training data to 5k, 50k, and 500k groups, to explore the impact of data scale in GDTs. As illustrated in Table 2, with the increase of the amount of training data, GDTs behave increasingly better in content consistency and prompt adherence. Interestingly, we find that FID would become lower when training on less data, which may be that it is easier to overfit to small datasets. We plan to further scale up our data to the level of hundreds of millions of groups in the future, in order to fully leverage the potential of GDTs.

### 4.2.3 GROUP SIZE

We gradually increase the upper limit of group size to 2, 4, and 8, and perform inference based on that limit. Note that doubling the group size will, in turn, double the sequence length in self-attention, leading to a corresponding increase in computational complexity, so we cap the maximum group size at 8 in our ablation. From the ablated results in Table 2, we find that larger group sizes lead to a more pronounced performance decline in image quality, content consistency, and prompt adherence. The reason may be that it is more difficult to learn the complex relationships across a large group of images. Besides, the scarcity of data of large group sizes prevents the model from being adequately trained. In the future, we would greatly scale our training data.

### 4.2.4 SDEDIT OR INPAINTING

When conditioned on a subset of the group data, using methods like SDEdit Meng et al. (2021) or trainable inpainting Xie et al. (2023); Xu et al. (2024), GDTs can be instructed to generate the remaining data of the group. Specifically, SDEdit is a training-free technique which provides the reference images that are added with the same noise step as the generated images during the denoising stage. In contrast, trainable inpainting concatenates the reference image to the noised one in channel dimension, allowing the model to "copy" the reference images and generate the remaining ones. In our ablation study, as illustrated in Table 2, it is observed that trainable inpainting performs better in image quality and content consistency, while the training-free SDEdit is good at prompt adherence. We adopt the model design of trainable inpainting in our GDTs.

### 4.2.5 QUALITY TUNING

While quality tuning is a common practice in visual generation models to enhance aesthetic appeal, we investigate its impact under the paradigm of group generation. As illustrated in Table 2, after the supervised fine-tuning on a small subset of high-quality image groups, GDTs exhibit significantly better image quality. We also find that quality tuning helps generating image groups with higher content consistency, while barely compromising the adherence to textual descriptions.

## 5 RELATED WORK

### 5.1 TEXT-TO-IMAGE GENERATION

The emergence of DDPM Ho et al. (2020) has catalyzed rapid advancements in text-to-image (T2I) generation. Earlier frameworks focused on T2I generation in pixel space, exemplified by GLIDE Nichol et al. (2022) and Imagen Saharia et al. (2022). In contrast, Stable Diffusion Rombach et al. (2022) introduced latent space for T2I generation, while DALLE-2 (unCLIP)Ramesh et al. (2022a) expanded this to a multimodal latent space. EMUDai et al. (2023) demonstrated that supervised fine-tuning on a small set of appealing images can significantly enhance generation quality. Unlike U-Net architectures, several approaches, including DiT Peebles & Xie (2023), Pixart Chen et al. (2023a), HunyuanDiT Li et al. (2024b), and SD3 Esser et al. (2024b), adopt transformers as their backbone.

### 5.2 CONTROLLABLE TEXT-TO-IMAGE GENERATION

**Personalization.** Personalization in T2I generation Cui et al. (2024); Salehi et al. (2024); Ham et al. (2024); Wang et al. (2024) aims to capture concepts like subject Li et al. (2023a); Kumari et al. (2023), person Xiao et al. (2023); Li et al. (2024a); Chen et al. (2024b; 2023b), style Liu et al. (2023a); Sohn et al. (2023), and image Ye et al. (2023b); Xu et al. (2023); Ramesh et al. (2022b). Techniques like Textual Inversion Gal et al. (2022) and DreamBooth Ruiz et al. (2022) facilitate concept embedding. Subject-driven methods Valevski et al. (2023); Chen et al. (2024b) use face recognition models for personalization.

**Spatial Control.** Spatial control in T2I generation Li et al. (2023b) is crucial for representing image structure. ControlNet Zhang et al. (2023) and UniControl Qin et al. (2023) are examples of models that incorporate positional signals for spatial control.

**Advanced Controllable Text-to-Image Generation.** New directions in controllable T2I generation include Attend-and-Excite Chefer et al. (2023), Composer Huang et al. (2023), Cocktail Hu et al. (2023), Cones Liu et al. (2023c), Universal Guidance Bansal et al. (2023), EMU2 Sun et al. (2024), and FreeDom Yu et al. (2023), which aim to enhance text alignment and achieve universal control.

### 5.3 GENERALIZATION ABILITY OF GENERATIVE MODELS

Beyond fundamental generative capabilities, recent approaches are investigating the generalization and versatility of models. ControlNeXt Peng et al. (2024) is designed to support both images and videos while incorporating diverse forms of control information. EMU2 Sun et al. (2024) demonstrates task-agnostic in-context learning capabilities. MT-Diffusion Chen et al. (2024a) achieves multi-modality diffusion through multi-task learning.

In contrast to the aforementioned methods, Group Diffusion Transformers aim to provide a general-purpose visual generation framework with the following capabilities: 1) no need for task-specific pretraining or finetuning; 2) generating multiple images in parallel; 3) conditioning on text or images; and 4) enabling zero-shot task generalization.

## 6 CONCLUSION AND LIMITATIONS

We reformulate most visual generation tasks into a **group generation** problem, thereby introducing a unified framework named **Group Diffusion Transformers** (GDTs). We present that with scalable, unsupervised, and task-agnostic pretraining on group data, GDTs could achieve competitive zero-shot performance on a vast array of visual generation tasks. Our results demonstrate the potential of GDTs as scalable, general-purpose visual generation systems.

Moreoever, we point out that there is still a discrepancy in image quality between GDTs and the state-of-the-art text-to-image models. The amount of group data for pretraining is also not sufficient yet, which has not fully unleashed the model's capabilities. We are optimistic that with an enlarged group dataset, we can further optimize the model's performance and reduce the discrepancy. In the future, we also plan to extend the time dimension of GDTs to enable multi-shot video generation, which can be naturally expressed under our group generation framework.

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

# A  APPENDIX

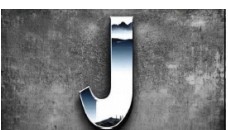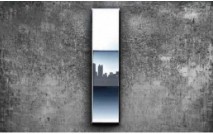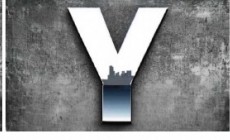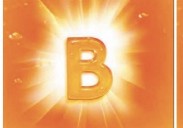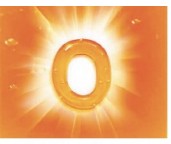

The image presents a large three-dimensional letter \"J\", made of polished metal with a smooth surface and a strong mirror reflection. The interior of the letter reflects a blurry cityscape or mountain range, primarily in dark blue and gray-white tones. The letter is placed against a rough gray cement wall, creating a sharp contrast that enhances the metallic texture and three-dimensionality.

The image is dominated by a large three-dimensional letter \"I\", made of polished metal with a mirror effect, reflecting the surrounding light. Inside the letter is a clear black and white buildings. The bottom is dark blue, contrasting with the bright white above. The background is a rough gray cement wall, creating visual tension with the metallic letter.

The image shows a large three-dimensional letter \"Y\", also made of polished metal with a mirror reflection. The letter reflects a relatively clear city outline; the buildings are arranged irregularly, and the overall color is dark. The bottom of the letter transitions to dark blue, contrasting with the gray cement wall in the background, creating an industrial and modern feel. The letter is three-dimensional and has a strong texture.

The main subject of the image is a transparent orange letter 'B', with a fine water droplet-like texture on its surface, giving it a crystal-clear texture. The letter 'B' is located in the center of the image, surrounded by bright orange rays of light radiating outwards, creating a dazzling light effect. Small orange droplets are scattered in the background, echoing the texture on the surface of the letter. The overall tone is warm, bright, and full of vitality.

The main subject of the image is a transparent orange letter 'O', with a fine water droplet-like texture on its surface, giving it a crystal-clear texture. The letter 'O' is located in the center of the image, surrounded by bright orange rays of light radiating outwards, creating a dazzling light effect. Small orange droplets are scattered in the background, echoing the texture on the surface of the number. The overall tone is warm, bright, and full of vitality.

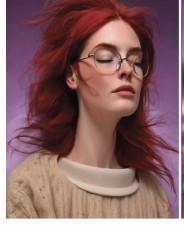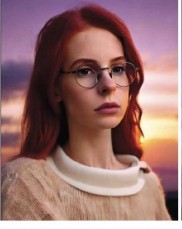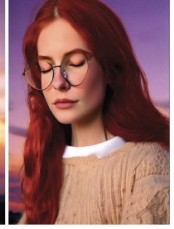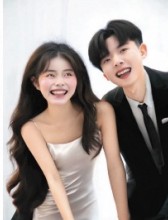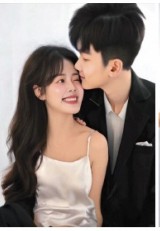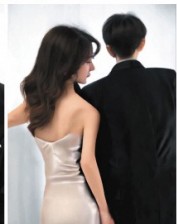

A young woman with long red hair is shown with her eyes closed, wearing a beige sweater with a white collar underneath and round glasses. The background features a sunset and a purple backdrop.

A young woman with long red hair is shown with her eyes closed, wearing a beige sweater with a white collar underneath and round glasses. The background features a purple backdrop, and her hair is blowing in the wind.

A young woman with long red hair is shown wearing a beige sweater with a white collar underneath and round glasses. She turns her head to look at the camera, with a sunset and a purple backdrop in the background.

In the photo, a young couple is all smiles. The girl is wearing a white slip dress and the boy is wearing a black suit with a white shirt and black tie. They are holding hands and look very sweet and happy.

In the photo, a young couple cuddling intimately. The girl is wearing a white slip dress and the boy is wearing a black suit. They are looking into each other's eyes and the girl has a sweet smile on her face, overflowing with happiness.

In the photo, a young couple is in an intimate pose. The girl is wearing a white slip dress with her back to the camera, revealing her bare back. The boy is wearing a black suits and looking at the girl's back affectionately.

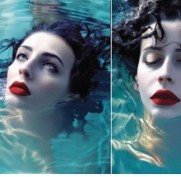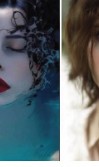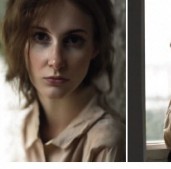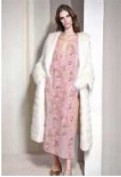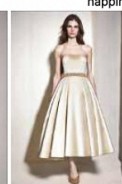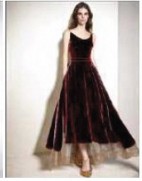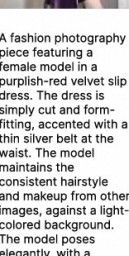

A young woman floats in the water, her head slightly lifted and her gaze directed upwards. Her dark hair spreads around her, contrasting sharply with her fair skin and vibrant red lipstick. The water surrounds her face and hair, creating a dreamlike atmosphere.

A young woman floats in the water with her eyes closed. Her dark hair spreads around her, contrasting with her fair skin and vibrant red lipstick. The water surrounds her face, creating a peaceful and serene atmosphere.

A young woman leans on a chest, gazing directly at the viewer. She has fair skin, her brown hair falls loosely around her shoulders, and her dark eyes hold a gentle expression. She wears a light-colored shirt, the neckline slightly open, revealing the graceful lines of her neck.

A young woman stands by the window, her head tilted slightly upwards as if feeling the caress of a gentle breeze. She has fair skin, her brown hair falls naturally, and her dark eyes are filled with hope for the future. She wears a light-colored shirt with loose, comfortable sleeves and dark pants, creating a simple and elegant look.

A fashion photography piece showcasing a female model in a pink printed maxi dress layered over with a long white furry coat. The model has long brown hair and sophisticated makeup; she stands against a light-colored backdrop in an elegant pose with her hands in her pockets. The dress has a V-neck design, is lightweight in texture, and features a soft print. The coat is fluffy and soft, with a high-end feel. The model is wearing gold strapped heels.

Fashion photography showcasing a female model in a champagne-colored strapless A-line dress. The dress is smooth in texture, with a strong drape, and an A-line skirt, cinched at the waist with a golden belt. The model's hairstyle and makeup are consistent with the previous image, and the background is similarly simple and light-colored. The model's pose is elegant, with her gaze directed forward. She wears gold heels that complement the dress.

A fashion photography image displaying a female model in a burgundy velvet maxi dress. The dress is a halter neck style, with a defined waist, and an asymmetric hemline, trailing on one side while revealing the ankle on the other. The dress is thick in texture, with a rich color, and the skirt flows slightly. The model's hairstyle and makeup remain consistent, against a simple light-colored background. The model's posture is elegant, and she wears gold high-heeled shoes.

A fashion photography piece featuring a female model in a purplish-red velvet slip dress. The dress is simply cut and form-fitting, accented with a thin silver belt at the waist. The model maintains the consistent hairstyle and makeup from other images, against a light-colored background. The model poses elegantly, with a confident smile. Her makeup is refined, with simple earrings. The model wears light gold heeled shoes.

Figure 1: **Detailed results of Group Diffusion Transformers.**

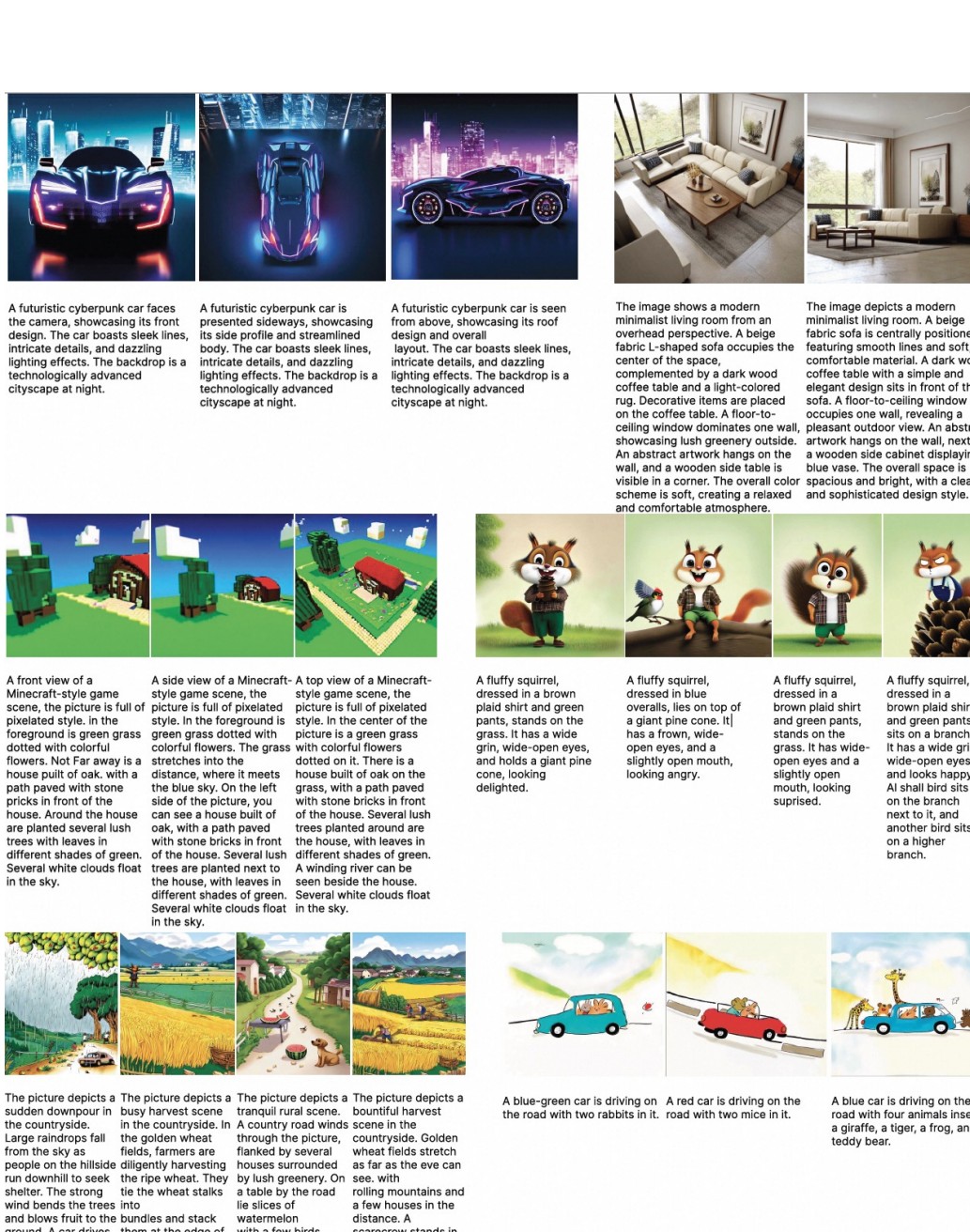

Figure 2: **Detailed results of Group Diffusion Transformers.**

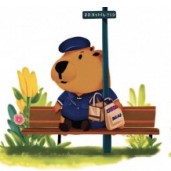 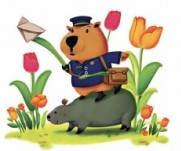 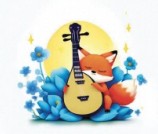 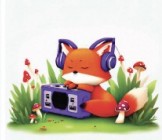 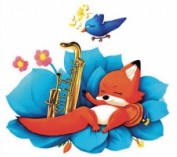

A cartoon capybara wearing a blue uniform and hat and carrying a mailbag sits on a bench next to a bus stop sign that reads ' BUS'. The capybara is surrounded by flowers and plants.

A cartoon capybara wearing a blue uniform and hat and carrying a mailbag sits on a bench next to a bus stop sign that reads ' BUS'. The capybara is surrounded by flowers and plants.

A cartoon capybara wearing a blue uniform and hat and carrying a mailbag stands atop a turtle walking through a garden of tulips and other plants. The capybara has a stem of a plant in its mouth, and an envelope files behind it.

An orange fox wearing headphones sits with its eyes closed on a grassy field, next to a purple vintage cassette player. Surrounding it are red mushrooms and plants bearing red fruits.

An orange fox is peacefully sleeping with its eyes closed in a blue flower bed, next to a golden saxophone. A blue bird is joyfully singing on top of the saxophone.

An orange fox with closed eyes snuggles next to a pipa, surrounded by blooming blue flowers, with a golden full moon and twinkling stars in the background.

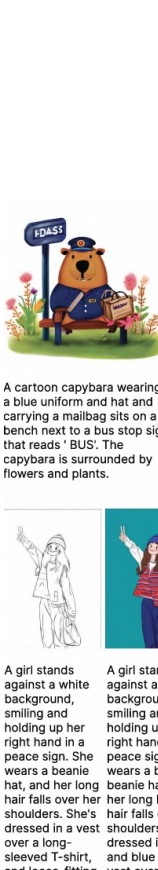 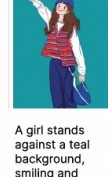 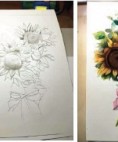 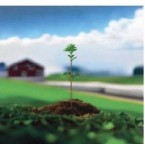 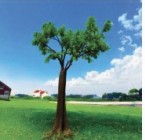 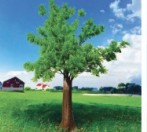 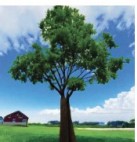

A girl stands against a white background, smiling and holding up her right hand in a peace sign. She wears a beanie hat, and her long hair falls over her shoulders. She's dressed in a vest over a long-sleeved T-shirt, and loose-fitting pants. A bag hangs from her shoulder, and sneakers complete her casual look.

A girl stands against a teal background, smiling and holding up her right hand in a peace sign. She wears a blue beanie hat, and her long brown hair falls over her shoulders. She's dressed in a red and blue striped vest over a white long-sleeved T-shirt, and loose-fitting blue jeans. A blue bag hangs from her shoulder, and white sneakers complete her casual look.

On a white sheet of paper, a pencil sketch of a bouquet of flowers. The bouquet consists of two blooming sunflowers and some green leavers and small flowers wrapped in wrapping paper and tied with a bow.

On a cream-colored sheet of paper, all areas of a bouquet are painted with watecolor paints. The bouquet consists of two blooming sunflowers and small flowers, wrapped in wrapping paper and tied with a bow.

On a sunny day, on a green grassland, several houses can be seen in the distance, with blue sky and white clouds dotted the sky. In the center of the screen, a small sapling has just been planted in the soil. It looks very fragile, but full of vitality.

On a sunny day, on a green grassland, several houses can be seen in the distance, with blue sky and white clouds dotted the sky. In the center of the screen, the once weak sapling has grown much taller, about 1 meter high, and its branches and leaves are even more lush.

On a sunny day, on a green grassland, several houses can be seen in the distance, with blue sky and white clouds dotted the sky. In the center of the screen, the tree has grown to 2 meters high, with lush branches and leaves, full of vitality.

On a sunny day, on a green grassland, several houses can be seen in the distance, with blue sky and white clouds dotted the sky. In the center of the screen, the tree has grown into a 3-meter-tall tall tree, with lush branches and leaves, covering the sky and sun. Standing west, it exudes strong vitality.

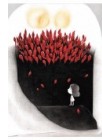 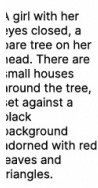 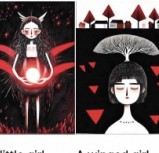 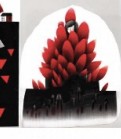 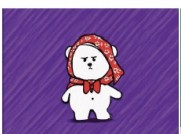 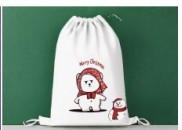 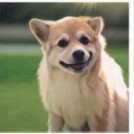 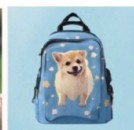

A girl with her eyes closed, a bare tree on her head. There are small houses around the tree, set against a black background adorned with red leaves and triangles.

A little girl standing on a cliff edge, looking up at a cluster of red plants with a glowing sphere in the center. Below the cliff is a black forest, with a white background speckled with red plants.

A winged girl holding a glowing sphere, surrounded by swirling red lines and stars. Her background is black, dotted with red plants.

A winged girl, mouth open, looking up at a cluster of red plants with a glowing sphere in the center. Beneath her are small houses and black plants.

A cartoon polar bear wearing a red floral headscarf and a red bow tie, with a serious expression, stands in the center of a purple striped background.

A white drawstring backpack with a cartoon polar bear wearing a red floral headscarf and a snowman with a red scarf and a carrot nose printed on it, with "Merry Christmas" written above.

The picture shows a cute little dog with fluffy fur, round eyes, and a lively nose, smiling kindly at the camera. The puppy's fur color is light brown, and its body posture appears very relaxed. Its limbs naturally hang on both sides of its body, and its tail gently sways, as if welcoming its owner's arrival. The background of the puppy is a green lawn, with sunlight shining through the clouds, illuminating the entire scene.

The picture shows a cute cartoon style backpack with a blue themed color and some cute patterns printed on it. On the front of the backpack, there is a photo of a small dog with fluffy fur, round eyes, and a lively nose. It is facing the camera with a friendly smile. The photo of the puppy is printed in the center of the backpack, with a moderate size that matches the overall style of the backpack. The background of the backpack is a bright yellow color, creating a lively atmosphere.

Figure 3: **Detailed results of Group Diffusion Transformers.**

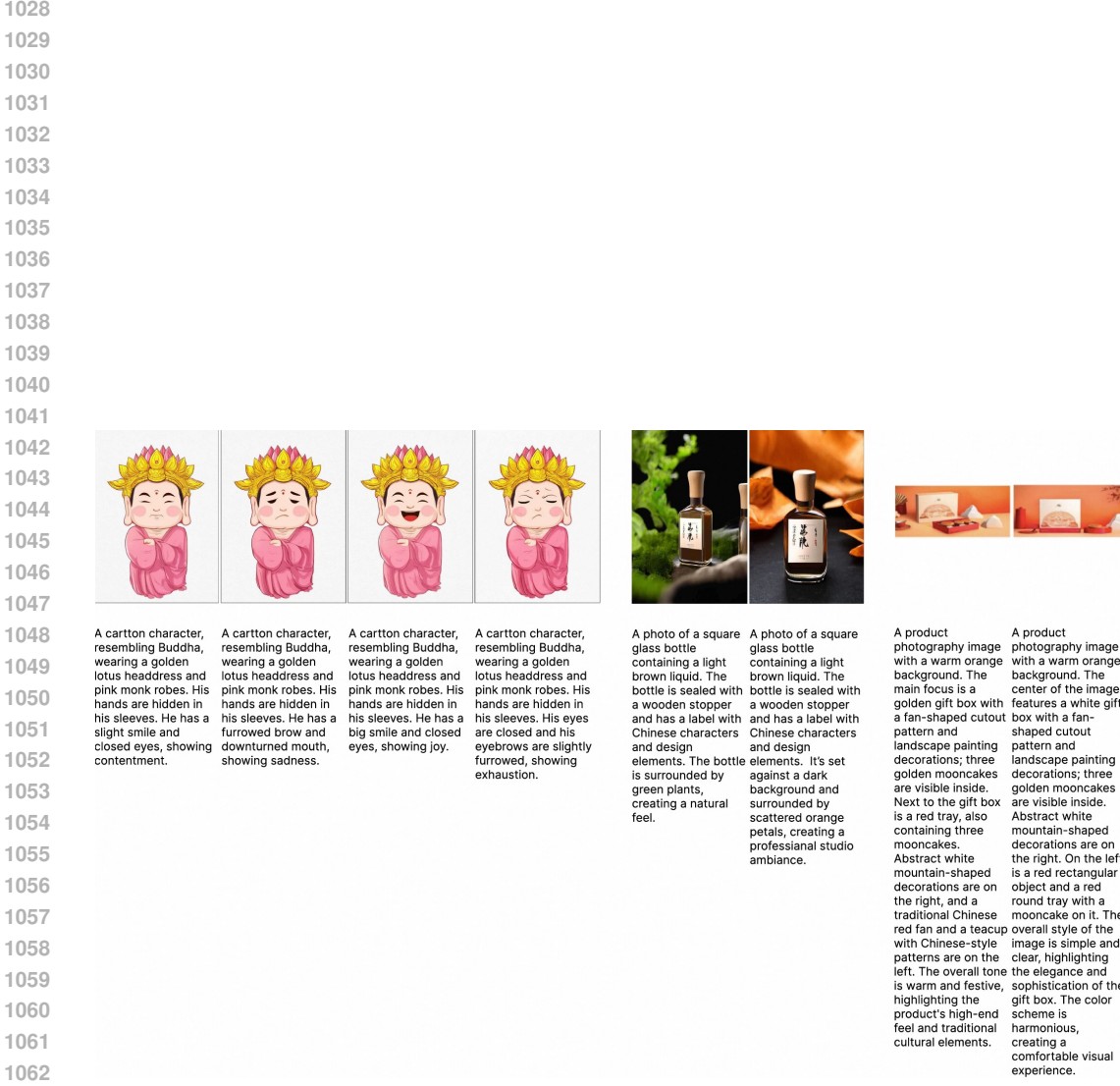

Figure 4: **Detailed results of Group Diffusion Transformers.**

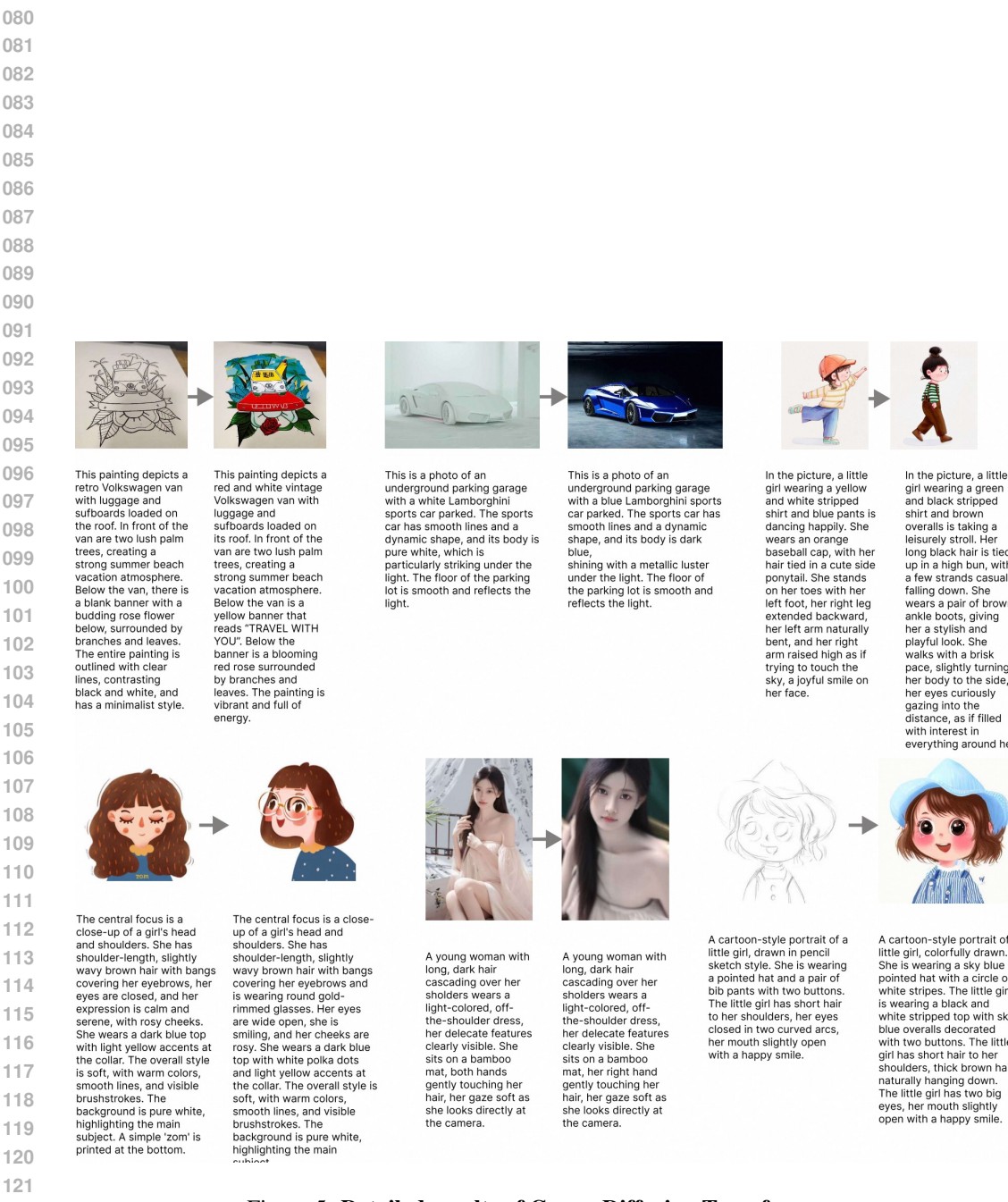

Figure 5: **Detailed results of Group Diffusion Transformers.**

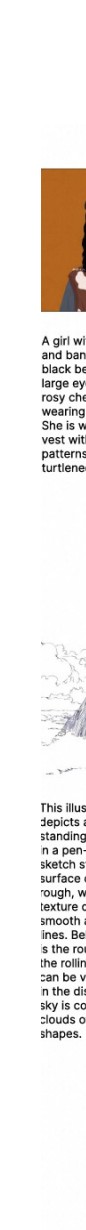
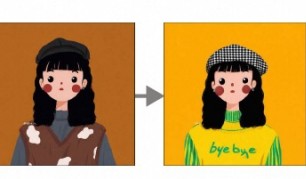
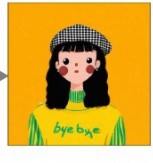
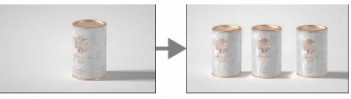
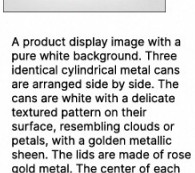
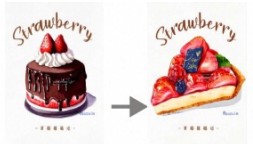

A girl with black bob hair and bangs is wearing a black beret. She has large eyes, a round face, rosy cheeks, and is wearing silver earrings. She is wearing a brown vest with white cloud patterns over a blue turtleneck.

A girl with black bob hair and bangs is wearing a black checkered beret. She has large eyes, a round face, rosy cheeks, and is wearing gold earrings. She is wearing a yellow sweater with the words 'byebye' printed on it and green striped collar and cuffs.

A product display image with a pure white background. A cylindrical metal can is placed in the center. The can is white with a delicate textured pattern on its surface, resembling clouds or petals, with a golden metallic sheen. The lid is made of rose gold metal. The center of the can features a golden brand logo, including the brand name \"PHYSICAL ART\" and the Chinese brand name \"悠然\". The bottom of the can is marked with \"CONCERT 45ML\". The overall style is simple, elegant, and refined.

A product display image with a pure white background. Three identical cylindrical metal cans are arranged side by side. The cans are white with a delicate textured pattern on their surface, resembling clouds or petals, with a golden metallic sheen. The lids are made of rose gold metal. The center of each can features a golden brand logo, including the brand name \"PHYSICAL ART\" and the Chinese brand name \"悠然\". The bottom of each can is marked with \"CONCERT 45ML\". The overall style is simple, elegant, and refined.

The center of the image is a chocolate cake. The bottom of the cake is red, the top is covered with thick chocolate sauce, and it's decorated with fluffy cream and two fresh strawberries. The cake rests on a dark base. Above the image is the title \"Strawberry\", with an elegant and flowing font. The background is simple, the overall style is fresh and sweet, creating a comfortable visual experience. The watercolor painting technique gives the image a soft color transition and a light texture.

The main subject of the image is a slice of strawberry tart. The tart crust is golden yellow, and it's topped with bright red strawberries, decorated with a few blueberries and cherries. There's a golden chocolate decoration on the strawberries. The cross-section of the tart shows a rich layering and the texture of the filling. Above the image is the title \"Strawberry\", consistent with the first image. The background is equally simple, and the watercolor painting technique creates a light and dreamy atmosphere.

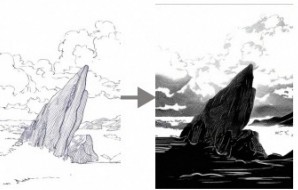
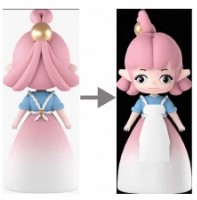

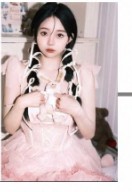
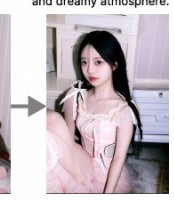

This illustration depicts a huge rock standing on the coast in a pen-and-ink sketch style. The surface of the rock is rough, with rich texture details and smooth and natural lines. Behind the rock is the rough sea, and the rolling mountains can be vaguely seen in the distance. The sky is covered with clouds of various shapes.

This illustration depicts a huge rock standing on the coast in the style of an engraving. The surface of the rock is rough, with rich texture details, and the use of dense lines to depict the effect of light and shadow. The bottom of the rock is beaten by the surging waves, and the the spray is splashing. In the distance are rolling mountains, and the sky is filled with clouds of various shapes, with sunlight shining through the clouds.

This is a 3D rendered image showcasing the back view of a cartoon girl. She has pink hair tied up in a high bun, adorned with a golden spherical ornament. She is wearing a blue top with a white apron tied around her waist, featuring rope-like detailing. Her dress is pink, gradually fading to white at the bottom. She has pointed ears, exhibiting an overall cute and sweet style.

This is a 3D rendered image showcasing the front view of a cartoon girl. She has pink hair tied up in a high bun, adorned with a golden spherical ornament. She has large eyes and cute pointed ears. She is wearing a blue short-sleeved top with a white apron. She is wearing a pink dress, gradually fading to white at the bottom. The overall style of the girl is cute and sweet.

A young Asian woman with long, black hair sits on the floor wearing a pink lace dress with white ribbon decorations tied in bows on her head. She holds a storybook and looks at the viewer with clear eyes. The room features white wood paneling and a white cabinet.

A young Asian woman with long, black hair sits on the floor wearing a pink lace dress with white ribbon decorations tied in bows on her head. She embraces a plush toy and looks gently ahead. The room features white wood paneling and a white cabinet.

A young Asian woman with long, black hair sits on the floor wearing a pink lace dress with white ribbon decorations tied in bows on her head. She rests one hand on her leg and looks directly at the viewer with clear eyes. The room features white wood paneling and a white cabinet.

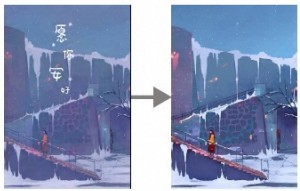
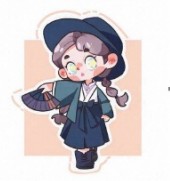
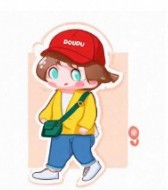

The scene presents a tranquil winter nightscape. Snow-laden stone buildings stand tall on either side, with dense icicles hanging from their roofs. A woman in a reddish-brown coat ascends a stone staircase, carrying a warm lantern. Dim yellowish lights emanate from the buildings, contrasting with the twinkling stars in the sky. Fine snowflakes fall from the sky, and a thin layer of snow covers the ground. The overall color palette is cool, creating a serene and peaceful atmosphere. In the distance, another figure can be vaguely seen standing at a building entrance. A lone, bare tree is visible in the lower right corner. Vertical Chinese characters reading "愿你安好" are displayed at the top.

The image depicts a cold winter night scene. Tall buildings on both sides are covered with thick snow and long icicles. A stone staircase winds upwards, with a figure in an orange-yellow coat walking along it. The buildings' lights are warm and yellowish, contrasting with the sparse stars in the night sky. Tiny snowflakes fall from the sky, and the ground is covered in snow. The color palette is cool, but the lights provide a touch of warmth, creating a quiet and slightly mysterious atmosphere. A bare tree and a partial view of the building interiors are visible in the background. The overall style is dreamy and slightly impressionistic.

The image presents an adorable chibi girl dressed in dark blue clothing. She wears a dark blue hat, a dark blue jacket, a white top underneath, a dark blue skirt, and black ankle boots. She has two braided pigtails and holds a dark purple folding fan. Her eyes are clear and bright, and her overall style is fashionable with traditional elements. The background is light orange with small dots around the edges, creating a fresh and cute style.

The image shows an adorable chibi girl in casual attire. She sports a red baseball cap with \"DOUDU\" printed on it, a yellow jacket over a white top, blue jeans, and white sneakers, along with a dark green crossbody bag. She has shoulder-length brown hair, slightly rosy cheeks, and a somewhat shy expression, creating a youthful and lively overall style. The background is light orange with small dots around the edges, maintaining a fresh and cute style.

Figure 6: **Detailed results of Group Diffusion Transformers.**

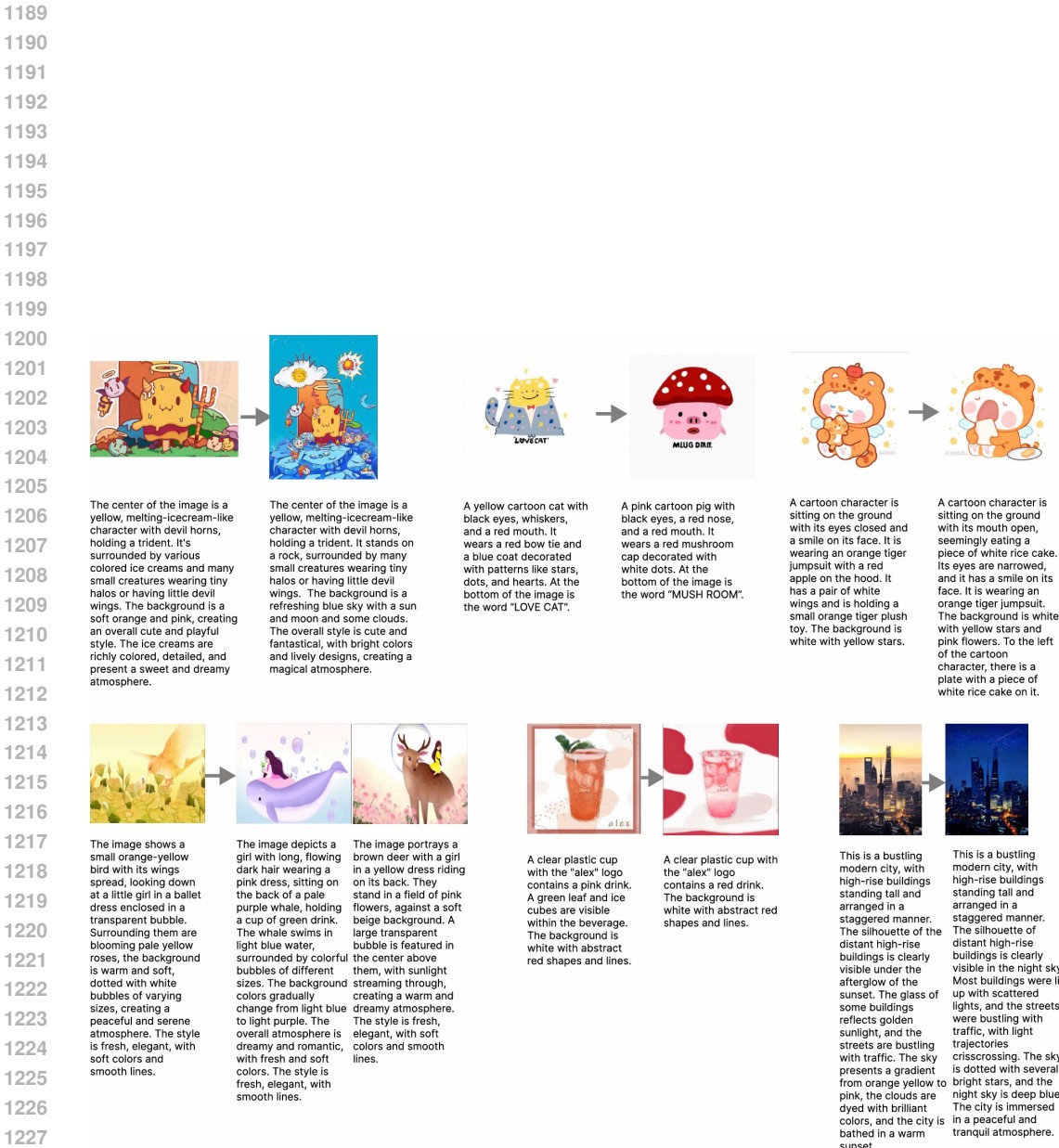

Figure 7: **Detailed results of Group Diffusion Transformers.**

