# OpenReview forum: "Group Diffusion Transformers are Unsupervised Multitask Learners"
_ICLR.cc/2025/Conference — ICLR 2025 Conference Withdrawn Submission_

### Official Review · Reviewer_QT22 · 2024-11-03

**Soundness:** 3
**Presentation:** 3
**Contribution:** 3
**Rating:** 6
**Confidence:** 3

**Summary:**

The author introduce Group Diffusion Transformers (GDTs), in which users can group-solve diverse visual tasks simultaneously (i.e. mainly generating multiple images). GDTs, sourced from SD3 & Pixart-Alpha, applies minor modifications on architecture, claimed to be a scalable method trained unsupervisely on task-agnostic data (i.e. image groups). Later at inference stage, GDTs can achieve zero-shot style transfer / sketching / colorization and many other visual tasks.

**Strengths:**

1. One of the major highlight of this work is providing an unsupervised “Group Training” of the specialized diffusion models using a series of correlated images and corresponding captions. The merit of such “Group Training” is that the latter trained model can be widely utilized in up to 30 downstream visual tasks.

2. Preparing a large-scale group date with correlated images and fine-grain prompt is valuable to the community. Not sure whether the author is be able to open-source it or not.

3. Despite the difficulty of evaluating around many visual task, the author did a reasonably effort providing numerical metrics (FID & CLIP score), evaluating the performance of the method, and ablation studies as well.

**Weaknesses:**

1. As the reviewer understands, the approach still relies heavily on a well-trained MLLM.

Specifically, the interface provided in paper session 2.5 involves first generating a group prompt using LLM and later using the group prompt to generate a series of correlated images.

2. Despite the almighty GDTs claimed by the author, the visualization of many tasks (Fig.1,2) shows that downstream tasks’ performance still has some issues. Explicitly speaking, using GDTs on image editing tasks, the quality of the results shows that preserving the ID/Style/Content remains challenging. The generated image is also not well-aligned with user input. This suggests GDTs will likely downgrade performance like many other all-in-one models.

**Questions:**

One major question is that whether the purposed GPTs can beat current SOTA methods on any of the visual tasks it claimed it can handle. As an example, can we compare GPTs with Instructional Pix2Pix on instructional based editing, and will GPTs win? I understands that it is infeasible (and unfair) for GPTs to beat SOTA on all claimed visual tasks, but the approach will be more persuasive if it can beat SOTA on some tasks. (And clearing my concern on Weaknesses No.2)

---

### Official Review · Reviewer_ipJh · 2024-11-03

**Soundness:** 3
**Presentation:** 3
**Contribution:** 2
**Rating:** 5
**Confidence:** 3

**Summary:**

The author introduces a novel framework called Group Diffusion Transformers (GDTs), which unifies diverse visual generation tasks by framing them as a group generation problem. In this approach, a set of images is generated simultaneously, optionally conditioned on a subset of the group. GDTs make minimal modifications by concatenating self-attention tokens across images. This group-based design supports scalability, unsupervised learning, and task-agnostic pretraining, achieving strong zero-shot performance in task evaluation settings.

**Strengths:**

- The presentation of the proposed method is clear and easy to understand.
- The proposed method, GDTs, is novel in its ability to generalize across various visual generation tasks, supporting scalable and task-agnostic pretraining.
- Experiments demonstrate that GDTs exhibit zero-shot capability across different visual generation tasks.
- Ablation studies indicate that GDTs have strong potential for scalable pretraining with larger datasets and a greater number of groups.

**Weaknesses:**

- The evaluation of GDTs performance is limited, as the user study results lack comparisons with other task-specific methods. While the evaluations of GDTs are intended to be zero-shot, including comparisons with task-specific methods would provide a clearer understanding of GDTs’ capabilities.
- The proposed benchmark is not fully utilized. It would be helpful to demonstrate GDT performance across different tasks within the benchmark to highlight its versatility.
- It is unclear whether pretraining in GDTs improves performance when fine-tuned for specific tasks. I suggest providing a few task-specific fine-tuning examples of GDTs, along with comparisons to existing task-specific methods on relevant benchmarks.

**Questions:**

See above weaknesses.

---

### Official Review · Reviewer_x6M5 · 2024-11-03

**Soundness:** 2
**Presentation:** 2
**Contribution:** 2
**Rating:** 3
**Confidence:** 4

**Summary:**

The paper presents group diffusion transformer (GDT) that can generate a group of coherent images via text prompt instructions. The technical contribution of the paper is rather simple. To allow the generation of group of coherent images, self-attention tokens across multiple images are concatenated. The paper explains how it is applied to different diffusion transformer architectures (e.g., PixArt that decouples self- and cross-attention, SD3 that couples self- and cross-attention). The important bit of a paper is their data collection. The training dataset is scrawled from the Internet that contains a group of images. After some manual and automatic processing, authors were able to collect 500K image groups and further refined to 10K image groups of higher quality. These image groups are then described using multimodal LLMs to capture correlations among the images within each group. In experimental results, authors provide two metrics -- user study, which rates the goodness of the results by the GDTs, and automated metrics that measures the FID, content consistency or prompt adherence.

**Strengths:**

* The paper tackles an important problem that extends the capability of text-to-image models to more diverse set of tasks without requiring specialized data for each new task.
* The authors proposed a simple technical solution that seems to work.
* I like the idea of solving the multiple visual generation and editing tasks in a unified framework.

**Weaknesses:**

* Overall, the paper lacks the comparison to existing works, making it difficult to evaluate how good the proposed method is. Though I understand that the paper might be the first of its kind that presents the unified solution for visual generation and editing tasks, there are many previous works that solve each of these problems, and having no comparison to those works makes the paper not very well grounded.

---
* Details on the training dataset is missing
  * How many tasks do training dataset cover and how do their distributions look like?
  * How many of these collected data are AI generated image sets?
  * How much manual refinement (if exists at all) is requires for generating descriptions for image sets?
  * See comments on zero-shot below.

---

* Several misleading claims:
  * It is misleading to say "unsupervised training on image groups". It is effectively equivalent to the "supervised training on input-output pair / tuples" in the context of this paper. As such, the title of the paper "unsupervised multitask learners" could be misleading. Supervision (or weak supervision at least) is required to construct a set of image groups.
  * Claims on zero-shot: authors should make sure that the training dataset do not contain the data for certain task to claim zero-shot. For example, can GDT perform subject customization if the training data does not contain a group of images that contain the same subject, or image completion from sketch if the dataset does not contain such (sketch, image) pairs? Also, would the data scale matter the zero-shot performance?

---

* Issue in evaluation
  * FID evaluation: while authors hypothesized that training on a small dataset is easier to overfit, thus providing lower FID, this is not particularly convincing. On the other hand it raises a question on the evaluation dataset used for FID calculation. Would love to hear more on the distribution of eval dataset and how overfitting on the small training dataset could result in lower FID. Also, it would be great if authors could provide some generated images from a model trained with different training data scales.
  * Unlike many other user study that evaluates the preference between the two or more images generated from different models, authors asked users to give a score from 1 to 5 for test examples and concluded that the average score of GDTs is higher than 3. However, this score is not calibrated at all, making it difficult to evaluate how good the model is.
  * Not all tasks considered in this paper is new and there are tasks where abundant previous works exist, such as subject / style customization, image colorization, image variation generation, etc. While I understand that the difference between the GDT and those models that solves a single task, but the quality matters. The proposed user preference study or ablation metrics does not provide any points of comparison to existing works, making it difficult to evaluate the quality of the GDT. Authors are required to provide at least a few additional data points for quality comparisons to existing works on more established tasks.

**Questions:**

Please see weakness for questions for clarification and additional results.

Misc:

* line 317. Stable Diffusion 3 citation is wrong.

**Details Of Ethics Concerns:**

* Authors have collected data from Internet but without providing any descriptions on the data filtering process due to copyright / license / terms of use.

---

### Official Review · Reviewer_2bza · 2024-11-09

**Soundness:** 2
**Presentation:** 3
**Contribution:** 3
**Rating:** 5
**Confidence:** 4

**Summary:**

This paper presents group diffusion transformer (GDT), targeting the problem of multi-image generation for multi-task learning. The key idea is to bind the self-attention layers in generative models such as Pixart. Experiments with user study and typical metrics on a newly established benchmark show that GDT is an effective multi-task learner.

**Strengths:**

- The proposed method is pretty simple and easy to follow.
- The paper is written clearly.

**Weaknesses:**

- **Evaluation**. The evaluation is only conducted on the authors' own benchmark, and no comparison to other methods is conducted. Besides, the metrics like CLIP-T are not that reliable. The authors are encouraged to conduct experiments on human-aligned image generation benchmarks.
- **Novelty**. While simple, I think the idea of binding multi-image attention is not quite novel. However, I would not say that such exploration should not be encouraged.
- **Solidness**. The authors are training the model on their own dataset. I am not sure if the effectiveness mainly comes from the dataset or the model design. The authors should try to train the model on some other datasets.

**Questions:**

na

---

### Note · Authors · 2024-11-18

I have read and agree with the venue's withdrawal policy on behalf of myself and my co-authors.